# Learning Pseudo 3D Guidance for View-consistent 3D Texturing with 2D Diffusion

## Abstract

Text-driven 3D texturing requires the generation of high-fidelity texture that conforms to given geometry and description. Recently, the high-quality text-to-image generation ability of 2D diffusion model has significantly promoted this task, by converting it into a texture optimization process guided by multi-view synthesized images. Thus the generation of high-quality and multi-view consistency images becomes the key issue. State-of-the-art methods introduce global consistency by treating novel view image generation as image inpainting conditioned on the texture generated by previously seen views. However, due to the error accumulation of inpainting itself and the occlusion between object parts, these inpainting-based methods often fail to deal with long-range texture consistency and the learned texture is of low quality. To address these, we present **P3G**, a text to 3D texturing approach based on learned **P**seudo **3**D **G**uidance. The key idea of P3D is to first learn a coarse but view-consistent texture, to serve as a semantics and layout guidance for high-quality view-consistent multi-view image generation. To this end, we propose a novel method to enable the learning of the pseudo 3D guidance, and design an efficient framework for high-quality and multi-view consistent image generation that incorporates both the depth map, the learned high-level semantics and layout guidance, and the previously generated texture. Quantitative and qualitative evaluation on variant 3D shapes demonstrates the superiority of our P3G on both consistency and quality.

## 1 Introduction

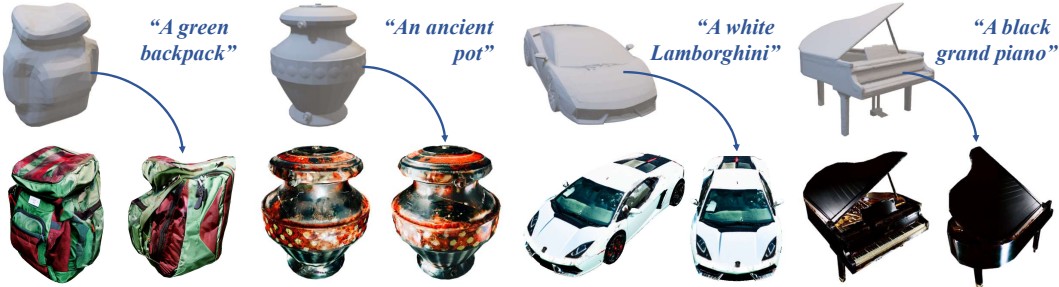

Figure 1: Given a text description and 3D mesh, our method can generate high-quality and view-consistent 3D textures.

High-quality 3D digital assets are crucial for applications such as virtual reality, gaming, and the movie industry. As a promising direction to greatly improve efficiency, automatic generation of 3D assets has aroused great interest and has been widely explored in computer graphics and computer vision. In this work, we focus on text-driven texturing of 3D meshes, which aims to generate high-quality texture of 3D meshes matching the given geometry and text description, as shown in Fig. 1.

Due to the lack of large-scale datasets of high-quality 3D assets and the corresponding text descriptions, most of the existing methods are built on large-scale 2D language-image models, such as CLIP (Radford et al., 2021) and diffusion models (Rombach et al., 2022), for realizing text-driven texture generation. Among them, the works of (Mohammad Khalid et al., 2022; Michel et al., 2022;

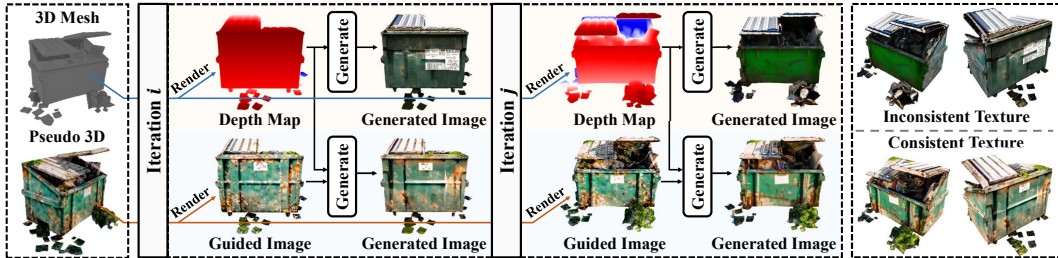

Figure 2: The learned textures without (the first row) and with (the second row) the pseudo 3D guidance. The pseudo 3D guidance can largely boost the view consistency.

Metzer et al., 2022) optimize the texture by maximizing the CLIP matching scores of the rendered 2D images and the input text. While the CLIP scores compare the high-level semantic consistency between text and images, their generated texture lacks details and suffers from global inconsistency (Richardson et al., 2023). Recently, with the help of photorealistic text-to-image generation ability of diffusion models (Ho et al., 2020), the visual quality of 3D texturing has been significantly promoted by optimizing texture using multi-view images generated by pre-trained 2D diffusion models. Although the quality of the image generated from a single view is impeccable, the consistency of images between different views is difficult to guarantee due to the natural randomness of the generation process, which significantly hurts the fidelity of the final texture. To tackle this problem, Richardson et al. (2023) and Chen et al. (2023) proposed to take previously generated texture into account when generating the image of a novel view through image inpainting. However, these inpainting-based methods often fail to ensure the long-range consistency of the whole object, since the occlusion between object parts in some views can be inevitable, and the inpainting may also bring errors which can accumulate as the number of views grows.

In this work, for the purpose of encouraging global consistency in the 2D image generation process of individual views, we propose a 3D guidance on high-level semantics and layout of the texture, which complies with the continuous characteristic of real 3D objects. Through this guidance, most of the randomness in high-level semantics and layout can be avoided and the remaining randomness in local details can be largely controlled by inpainting, therefore ensuring a global consistency of the generated 3D texture when observed from different views. The guidance is called pseudo 3D guidance, which is a rough 3D texture learnt by a 2D generative model utilizing the property that the 2D rendering at any view of a 3D object obeys the real image distribution. Although it is a pseudo one without high-quality details, the high-level semantics and layout are sufficient for guiding the 2D diffusion model to complement such details. The learning of the pseudo 3D guidance and the guided detail generation forms our method, namely, learning **P**seudo **3**D **G**uidance for view-consistent 3D texturing with 2D diffusion, abbreviated as **P3G**.

Specifically, our P3G is two-stage. In the first stage, for generating the pseudo 3D guidance with view-consistent high-level semantics and layout, we set as the optimization objective that any view-specific image sample to be in the high density area of the image distribution described by a diffusion model, which is implemented by score distillation sampling (SDS) (Poole et al., 2022), gradually updating a randomly initialized texture via constraining all of its 2D view renderings. Based on this key idea, we design a depth-based SDS to introduce geometric constraint, an optimization strategy from latent space to RGB space to improve visual quality, and a neural field-based texture model to enable convenient switching from latent space to RGB space. The semantics and layout guidance derived from the pseudo 3D object is implemented by rendering 2D images from it as the intermediate results of the diffusion model's denoising process, based on the observation that the denoising operation are good at refining details for generating high-quality 2D images. Moreover, we design a view selection strategy to do the detail refinement with small distortion from as few views as possible, by estimating non-distorted area covered by each view.

With above framework, on the one hand, the visual quality is guaranteed by the powerful text-to-image diffusion model. On the other hand, the randomness of each generation is controlled through the pseudo 3D guidance and the inpainting operation, thus achieving globally consistent texturing. We conduct extensive comparisons on variant 3D shapes with previous methods to demonstrate the effectiveness of our P3G. Quantitative, qualitative evaluation and user study shows that our P3G improves the consistency while maintaining the powerful generation ability of 2D diffusion model.

## 2 RELATED WORK

We study text-driven texturing of 3D shapes. There exist various 3D representations in computer vision and computer graphics (Shi et al., 2022), including point clouds, voxel grids, neural implicit functions, and meshes. Among them, polygonal mesh is primarily used in existing 3D production pipelines and we will deploy our method based on it. Our method is related to both text-to-image generation and text-to-texture generation.

**Text-to-Image Generation.** Text-driven image synthesis is a long-standing problem in computer vision and computer graphics. Recently, the large-scale vision-language dataset, such as LAION (Schuhmann et al., 2022; 2021), has driven rapid progress in text-to-image understanding and generation. For example, the state-of-the-art Stable Diffusion (Rombach et al., 2022) that trained on LAION-5B can generate high-quality 2D images. By conditioning the diffusion model on CLIP text embedding, it can generate impressive images according to the text description. Later, more and more extensions are realized to control pre-trained diffusion models to support additional input conditions, such as depth maps, edge maps, key points, etc.

**Text-to-Texture Generation.** Compared to 2D image generation, text-driven texture generation of 3D shapes is much more complicated, and requires attention to both shapes and text description. While early works adopt probabilistic models or study geometric texture synthesis for some specific categories (De Bonet, 1997; Efros & Leung, 1999; Aneja et al., 2023), recent advances explore data-driven approaches for zero-shot text-driven texturing of 3D shapes. Yet, unlike the massive text-to-image datasets, high-fidelity 3D data is relatively scarce. This has inspired several works to explore text-driven 3D texture generation with pre-trained 2D text-to-image models.

For instance, CLIP-Mesh (Mohammad Khalid et al., 2022) and Text2Mesh (Michel et al., 2022) utilize CLIP matching scores as criteria for texture optimization. Yet the CLIP scores compare the high-level semantic consistency between text and images, their generated texture is of low quality and lacks details (Richardson et al., 2023). For better visual quality, recent works explore pre-trained 2D text-to-image generation models for 3D texture generation (Metzer et al., 2022; Poole et al., 2022; Chen et al., 2023). The pioneering work of TEXTure (Richardson et al., 2023) projects the high-quality 2D images generated by the 2D diffusion model (Rombach et al., 2022) back to the mesh vertices. To cover the entire 3D mesh, it iteratively generates 2D images under different viewpoints. However, due to the stochastic nature of the generation process and the inevitable occlusions between object parts, this iterative inpainting strategy suffers from view inconsistency.

It is worth noting that another branch of texture generation methods are based on training a generative model using specific 3D datasets (Oechsle et al., 2019; Siddiqui et al., 2022; Yu et al., 2023b). Due to the limited 3D data and the difficulty of 3D texture representation, these methods can only be applied to specific classes, thus lossing the ability to match input text. And the textures they generate are relatively simple due to the quality of the dataset. In addition, some works are also trying to generate both 3D shape and texture given text prompt (Poole et al., 2022; Lin et al., 2023; Jain et al., 2022; Xu et al., 2023; Li et al., 2022). For example, the representative method DreamFusion (Poole et al., 2022) directly optimizes a neural radiance field (NeRF) with the SDS loss. Yet, it has no extra constraint on the learned shape and cannot be used for texture generation for a given mesh.

## 3 METHOD

### 3.1 LEARNING PSEUDO 3D GUIDANCE

The overall pipeline of the pseudo 3D guidance module is given in Figure 3. The pseudo 3D guidance is realized by learning a view-consistent coarse texture from 2D image generation model. Specially, the texture is generated in a multi-view optimization manner. Starting from random color, the texture is updated iteratively from random viewpoints by score distillation sampling (SDS) (Poole et al., 2022), based on the ability of the diffusion model to update random samples toward high probability areas. Since the optimization objective requires that the 2D images rendered at any viewpoint are in high probability areas in the natural image distribution described by the diffusion model, the final texture will be consistent. The learning of the pseudo 3d guidance consists of depth-guided SDS, optimization in latent and RGB space, and texture modeling.

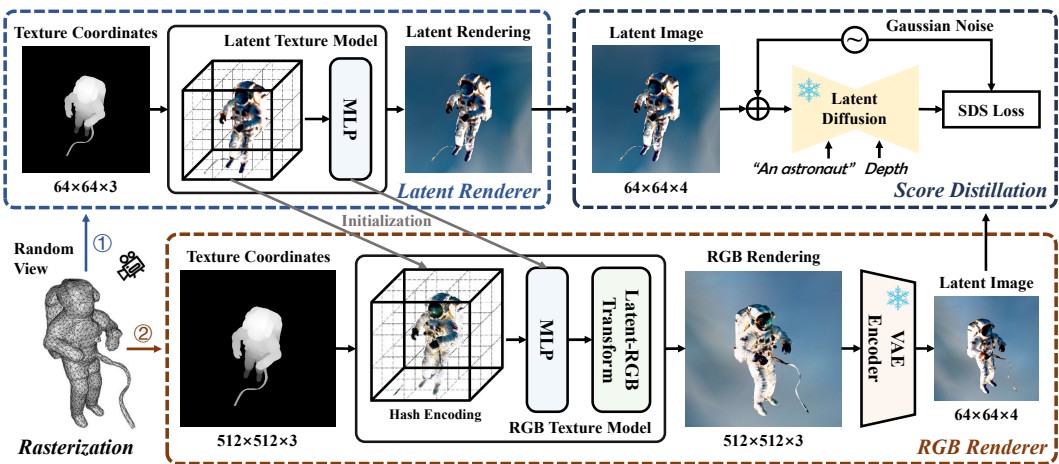

Figure 3: The pipeline of the pseudo 3D guidance learning module.

**Depth-guided SDS.** The SDS is originally proposed for text-to-3D generation and there is no geometry constraint. For texture synthesis, however, we find that directly applying the original SDS sometimes causes mismatching of the generated texture and the geometry, because the SDS requires a large classifier-free guidance weight (Ho & Salimans, 2021) that harms the association between the generated texture and the geometry. To solve this problem, we opt for a depth-based denoiser $\mathcal{F}_{depth}(\boldsymbol{x}_t; t, \boldsymbol{c}, \boldsymbol{d})$ which takes the geometry into account by conditioning on the depth map $d$. In each iteration, we first render an image $\boldsymbol{x}$ and the corresponding depth map $\boldsymbol{d}$ from a randomly sampled viewpoint $\boldsymbol{v}$ through a differentiable renderer $\mathcal{G}_{\boldsymbol{\theta}}$. The renderer $\mathcal{G}_{\boldsymbol{\theta}}$ contains a texture model with parameters $\boldsymbol{\theta}$, which are randomly initialized and optimized to consistent and realistic textures. The viewpoint $\boldsymbol{v}$ is defined by the azimuth angle $\alpha$, elevation angle $\beta$, and the radius $r$ of the camera. Then a time step $t$ is sampled and a random noise $\epsilon$ is added to $\boldsymbol{x}$ according to the diffusion process

$$\boldsymbol{x}_t = \sqrt{\alpha_t}\boldsymbol{x} + \sqrt{1 - \alpha_t}\epsilon, \tag{1}$$

and the gradient of $\boldsymbol{x}$ conditioned on the text and the geometry is calculated by

$$\nabla_{\boldsymbol{x}}\mathcal{L}_{SDS} = w(t)(\mathcal{F}_{depth}(\boldsymbol{x}_t; t, \boldsymbol{c}, \boldsymbol{d}) - \epsilon), \tag{2}$$

Finally the texture model is updated by the gradient *w.r.t.* $\boldsymbol{\theta}$

$$\nabla_{\boldsymbol{\theta}}\mathcal{L}_{SDS} = \nabla_{\boldsymbol{x}}\mathcal{L}_{SDS} \cdot \frac{\partial \boldsymbol{x}}{\partial \boldsymbol{\theta}}, \quad \boldsymbol{\theta} = \boldsymbol{\theta} - \gamma \cdot \nabla_{\boldsymbol{\theta}}\mathcal{L}_{SDS}, \tag{3}$$

where $\partial \boldsymbol{x}/\partial \boldsymbol{\theta}$ is calculated through the differentiable renderer and $\gamma$ is the learning rate.

**Texture Optimization in Latent Space.** For implementation, we use the open source Stable Diffusion (Rombach et al., 2022) as the denoiser, which uses a variational autoencoder (VAE) to project an RGB image $\boldsymbol{x}^{rgb} \in \mathbb{R}^{h \times w \times 3}$ to a latent image $\boldsymbol{x}^{lat} \in \mathbb{R}^{\frac{h}{8} \times \frac{w}{8} \times 4}$ and processes in the latent space. Considering this property, we conduct SDS in the latent space in the early stage of optimization, since the gradient *w.r.t.* the latent image can be obtained directly from the diffusion model and is more stable. Specifically, we first initialize the texture model to represent latent color and render latent images, and the gradient *w.r.t* the texture model becomes

$$\nabla_{\boldsymbol{x}^{lat}}\mathcal{L}_{SDS} = w(t)(\mathcal{F}_{depth}(\boldsymbol{x}_t^{lat}; t, \boldsymbol{c}, \boldsymbol{d}) - \epsilon), \quad \nabla_{\boldsymbol{\theta}}\mathcal{L}_{SDS} = \nabla_{\boldsymbol{x}^{lat}}\mathcal{L}_{SDS} \cdot \frac{\partial \boldsymbol{x}^{lat}}{\partial \boldsymbol{\theta}}. \tag{4}$$

**Texture Optimization in RGB Space.** Although a stable texture can be obtained through latent space optimization, its resolution is limited due to the low resolution of the latent image. In order to improve the sharpness of the texture as much as possible and provide a good reference for the fine stage, we further bring the SDS to the high-resolution RGB space.

We first convert the latent texture model to an RGB one by applying a point-wise color projection (Metzer et al., 2022) which transforms the four-dimensional latent color to three-dimensional RGB color, and render high-resolution RGB images from the converted texture model. For calculating the gradient *w.r.t.* the RGB image, we first use the VAE encoder $\mathcal{E}$ to convert it to a latent image

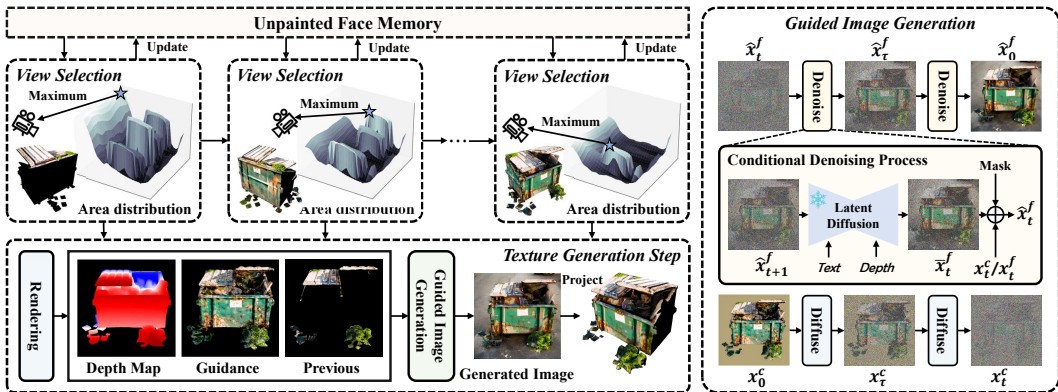

Figure 4: The pipeline of the high-quality texture generation module.

as $\boldsymbol{x}^{lat} = \mathcal{E}(\boldsymbol{x}^{rgb})$, and use the SDS to obtain the gradient *w.r.t.* the latent image. So in the RGB optimization, the gradient *w.r.t* the texture model becomes

$$\nabla_{\boldsymbol{x}^{lat}}\mathcal{L}_{SDS} = w(t)(\mathcal{F}_{depth}(\boldsymbol{x}_t^{lat}; t, \boldsymbol{c}, \boldsymbol{d}) - \epsilon), \quad \nabla_{\boldsymbol{\theta}}\mathcal{L}_{SDS} = \nabla_{\boldsymbol{x}^{lat}}\mathcal{L}_{SDS} \cdot \frac{\partial \boldsymbol{x}^{lat}}{\partial \boldsymbol{x}^{rgb}} \cdot \frac{\partial \boldsymbol{x}^{rgb}}{\partial \boldsymbol{\theta}}, \quad (5)$$

where the Jacobian $\partial \boldsymbol{x}^{lat}/\partial \boldsymbol{x}^{rgb}$ is calculated by the encoder $\mathcal{E}$ and $\partial \boldsymbol{x}^{rgb}/\partial \boldsymbol{\theta}$ is calculated by the differentiable renderer.

**Texture Model Construction.** Our optimization framework from latent space to RGB space requires the texture model to have two properties. Firstly, the texture model should be easy to convert from latent color to RGB color, and adapt the resolution change. Secondly, it should have smoothness in the 3D space, since the gradient from the SDS loss is not always stable. Considering these factors, we instantiate the texture model as a multiresolution hash encoding (Müller et al., 2022) which encodes the feature of each point in the 3D space via interpolation the features on multiresolution grids, and a multilayer perceptron (MLP) which project multiresolution features to colors.

In short, given a 3D coordinate, the texture model returns the color at it. For rendering, we first obtain the depth and 3D coordinate of each pixel in the 2D image using differentiable rasterization (Laine et al., 2020), and use the texture model to transform the coordinates to colored image. Since the texture model encodes continuous colors in the 3D space, the output resolution can be controlled only by rasterization, and the convention from latent space to RGB space can be simply achieved by applying a linear transformation to the latent texture model. And the smoothness is ensured by the interpolation operation, which further stabilizes texture updates via the SDS.

### 3.2 HIGH-QUALITY TEXTURE GENERATION

After learning the pseudo 3D guidance, we conduct high-fidelity view-consistent texture generation based on it. The process is illustrated in Figure 4. We iteratively generate 2D images under different views and complete the underlying 3D texture through inverse rendering. The key is high-fidelity and multi-view consistent image generation. To realize this, as shown in Figure 4, in each iteration, we dynamically select viewpoints that contain more ungenerated textures, and generate an image by considering the depth map, the high-level semantics and layout from the pseudo 3D guidance, and the previously generated texture, in a reasonable manner. There are three main modules for the high-fidelity texture generation process, as follows,

**View-adaptive Texture Completion.** With the photorealistic image generation ability of 2D diffusion model, we gradually complete the 3D texture by generating images from different views. In this paradigm, one of the key issue is how to deal with overlapping part of the texture under different views. To resolve this conflict, given a rendering image from the $i$-th view, we define the texture region that this view is responsible for by pixels that satisfy

$$\cos\langle\vec{\boldsymbol{n}}, \vec{\boldsymbol{s}}_i\rangle > \max_p \cos\langle\vec{\boldsymbol{n}}, \vec{\boldsymbol{s}}_p\rangle, \ p \in \{1, 2, \ldots, i-1\}, \quad (6)$$

where $\vec{\boldsymbol{n}}$ is the normal of the corresponding face and $\vec{\boldsymbol{s}}_i$ is the sight direction of this view. This strategy encourages each of the texture to be updated at the best possible view where the corresponding

face is parallel to the imaging plane, reducing the effects of distortion when projecting 3D faces to 2D. The corresponding 2D image region to be generated is recorded by a binary mask $\boldsymbol{m}_{inpaint}$. After image generation, the masked region is projected back to the texture by inverse rendering, to update the corresponding texture.

**Efficient View Selection Strategy.** Another core module is the selection and generative order of views during the multi-view image generation. We aim to cover the texture using as few views as possible, while maintaining the requirement that every part of the texture is generated at a relatively good view. Therefore, we first define what a relatively good view is, and then dynamically select a view that covers as many textures to be generated as possible by estimating the distribution of faces that can be updated well under different views.

Specifically, we restrict the camera to be on a sphere with a fixed radius centered on the target, and directed towards the center of the sphere. Then the view is determined by two parameters, *i.e.*, the azimuth angle $\alpha$ and the elevation angle $\beta$. For each view $i$, we define its covered texture area as

$$A_i = \sum_{f \in \mathbb{F}_k} a_f, \ \mathbb{F}_k = \{f | f \in \mathbb{U}_j \wedge \cos\langle \vec{\boldsymbol{n}}_f, \vec{\boldsymbol{s}}_i \rangle < \delta\}, \tag{7}$$

where $a_f$ is the area of the $f$-th face in the 3D mesh, $\mathbb{U}_k$ is the unpainted faces at iteration $k$, and $\delta$ is a threshold to limit the statistics on faces with smaller distortion. For the $k$-th texture generation iteration, we select a view by $\arg\max_i A_i$, and update $\mathbb{U}$ by subtracting the updated faces.

**View-Consistent Image Generation.** Due to the natural randomness of the generation process, the images generated from different views will be inconsistent and geometrically mismatched if no control signal is introduced. As for geometry matching, we realize it by adopting a depth-based diffusion model $\mathcal{F}_{depth}(\boldsymbol{x}_t; t, \boldsymbol{c}, \boldsymbol{d})$ as mentioned above.

To enhance view consistency, we propose to generate images conditioned on the consistent coarse texture learned by the pseudo 3D guidance module. To achieve this with an off-the-shelf diffusion model, we treat the coarse texture as a layout and semantics guidance of the final image. Then based on the phenomena that the early and middle stages of the denoising process focus on image layout and semantics (Meng et al., 2021; Yu et al., 2023a), we hijack the denoising process at time step $\tau$ of the late stage by setting the foreground region to $\boldsymbol{x}_\tau^c$, which is obtained by adding random noise to the image $\boldsymbol{x}^c$ rendered from the coarse texture. Through the consistent high-level semantics and layout guidance, most of the randomness during the generation could be avoided.

Further, to avoid the potential inconsistency in local details induced by the diffusion denoising process, we additionally consider the inpainting operation that facilitates the consistency between the newly generated regions and the previous ones. Let the region to be generated is indicated by a binary mask $\boldsymbol{m}_{inpaint}$, inpainting is achieved by setting the unmasked region to the reference image obtained from previously learned texture at the end of each denoising step (Lugmayr et al., 2022). Considering that this general pre-trained diffusion-based inpainting strategy cannot address the long-range consistency very well (Richardson et al., 2023), we further introduce an additional inpainting-specific denoiser $\mathcal{F}_{inpaint}(\boldsymbol{x}_t; t, \boldsymbol{c}, \boldsymbol{m}_{inpaint})$ to complement the view-consistent texture for the masked region. These two inpainting operators complement each other as the $\mathcal{F}_{inpaint}(\boldsymbol{x}_t; t, \boldsymbol{c}, \boldsymbol{m}_{inpaint})$ ignores the depth guidance and may generate geometrically inconsistent textures. In summary, starting from a Gaussian noise $\hat{\boldsymbol{x}}_T^f$, where $T = 1000$ is the maximum time step specified by the pretrained diffusion model, the modified conditional denoising process can be described as

$$\tilde{\boldsymbol{x}}_{t-1}^f = \begin{cases} \mathcal{F}_{inpaint}(\hat{\boldsymbol{x}}_t^f; t, \boldsymbol{c}, \boldsymbol{m}_{inpaint}), & \tau < t < 800 \wedge t \bmod 2 = 1, \\ \mathcal{F}_{depth}(\hat{\boldsymbol{x}}_t^f; t, \boldsymbol{c}, \boldsymbol{d}), & otherwise, \end{cases} \tag{8}$$

where $\hat{\boldsymbol{x}}_t^f$ is the predicted image in the fine stage at time step $t$, and for each denoising step

$$\hat{\boldsymbol{x}}_{t-1}^f = \begin{cases} \boldsymbol{m}_{object} \odot \boldsymbol{x}_{t-1}^c + (1 - \boldsymbol{m}_{object}) \odot \tilde{\boldsymbol{x}}_{t-1}^f, & t \leq \tau, \\ \boldsymbol{m}_{inpaint} \odot \tilde{\boldsymbol{x}}_{t-1}^f + (1 - \boldsymbol{m}_{inpaint}) \odot \boldsymbol{x}_{t-1}^f, & t > \tau, \end{cases} \tag{9}$$

where $\boldsymbol{m}_{object}$ is the foreground mask, $\boldsymbol{x}_t^c$ and $\boldsymbol{x}_t^f$ are obtained by adding noise to image $\boldsymbol{x}^c$ rendered from coarse texture and image $\boldsymbol{x}^f$ rendered from already generated fine texture, respectively. Please

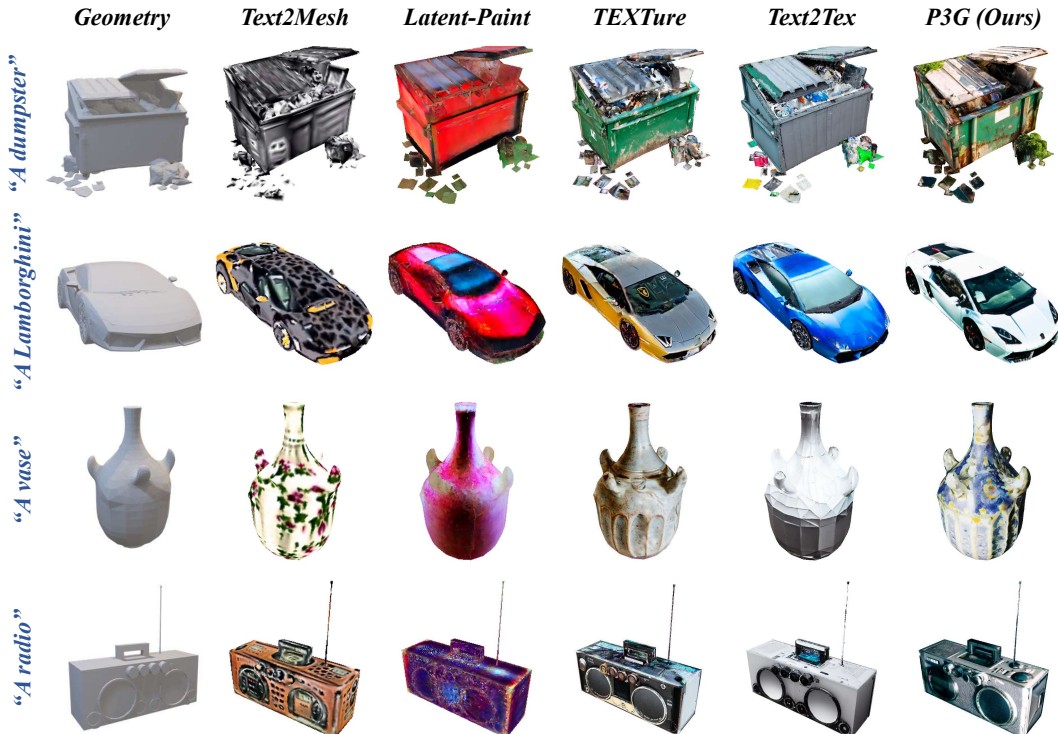

Figure 5: Visual comparisons of text-driven 3D texturing. For each row, the leftmost plot shows the input text and 3D mesh. We present rendering images under the same view for different methods.

note that in Eq. 9, for time step $t < \tau$, we fix the texture *w.r.t.* the foreground object region to the coarse texture and only update the background texture to better maintain the high-level semantics and layout of the pseudo 3D guidance.

To summarize, the high-fidelity texture generation stage is performed in an iterative manner. Initializing $\mathbb{U}$ to all faces of the 3D mesh, we iteratively select view $(\alpha, \beta)$, update the unpainted face set $\mathbb{U}$, generate a high-quality image from this view conditioned on the geometry, the texture from the coarse stage, and the previously generated texture, then project the image to the texture. The iteration terminates when a preset number of times is reached, or when there is very little content to be generated on a view. In this process, the view selection strategy reduces the number of views required and encourages contiguous areas to be updated. Meanwhile, the view-consistent image generation module helps to ensure global consistency via the conditional generation strategy.

# 4 EXPERIMENTS

## 4.1 IMPLEMENTATION DETAILS

**Pseudo 3D Guidance Learning Stage.** We use the depth-conditioned Stable Diffusion (Rombach et al., 2022) (SD-v2-depth). For both latent space and RGB space, the optimization is performed by mini-batch with a batch size of 4 for 500 iterations using the Adam (Kingma & Ba, 2014) optimizer with a learning rate of 0.01. We adopt nvdiffrast (Laine et al., 2020) for rendering due to its efficiency. For the latent space optimization, we set the rendering resolution to $64 \times 64$ and the time step $t$ is sampled from $[20, 980]$. For the RGB space optimization, the rendering resolution is set to $512 \times 512$ and the time step $t$ is sampled from a smaller range $[20, 500]$. We add a linear transformation as suggested in (Metzer et al., 2022) on the hash encoding and MLP optimized in latent space and use sigmoid activation to limit the RGB value range.

**High-quality Texture Generation Stage.** We use the same depth-conditioned diffusion model as in the coarse stage, and an extra inpainting model SD-v2-inpainting. The threshold $\delta$, the initial time step $\tau$, and the maximum number of views are set to $\cos 30°$, 500, and 10, respectively. For

| Method | CLIP Score ↑ | CLIP-IQA ↑ | CLIP Variance ↑ |
|---|---|---|---|
| Text2Mesh (Michel et al., 2022) CVPR22 | **26.33** | 30.82 | 91.56 |
| Latent-Paint (Metzer et al., 2022) CVPR23 | 24.19 | 28.88 | **91.59** |
| TEXTure (Richardson et al., 2023) SIGRAPH23 | 24.73 | 43.96 | 90.46 |
| Text2Tex (Chen et al., 2023) ICCV23 | 23.48 | 40.92 | 90.87 |
| P3G (*Ours*) | 24.56 | **44.22** | 90.83 |

Table 1: Numerical comparisons of different methods. The best result is highlighted in bold.

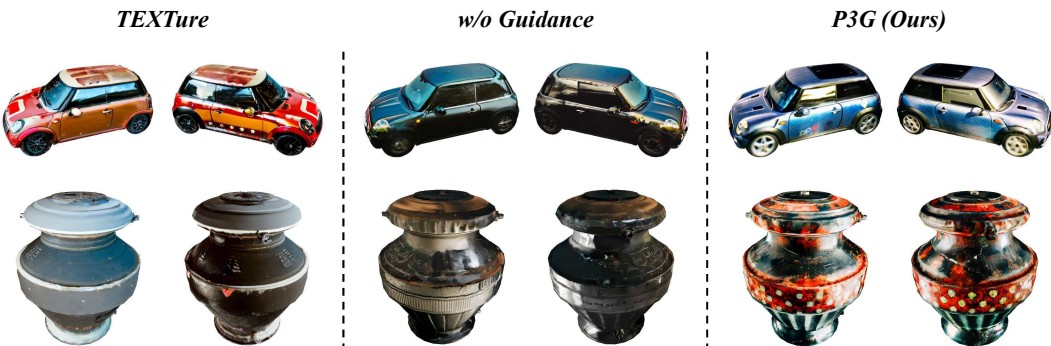

Figure 6: Ablation study on pseudo 3D guidance.

rendering, the texture is represented as an atlas through UV mapping calculated by Xatlas (Young, 2020), and kaolin (Fuji Tsang et al., 2022) is used as the renderer for its adaptability to texture atlas.

## 4.2 MAIN RESULTS

**Dataset.** We collect about 50 multi-category meshes from ModelNet (Wu et al., 2015), ShapeNet (Chang et al., 2015), and some open source projects (Michel et al., 2022; Richardson et al., 2023) for evaluation. Each mesh is preprocessed using ManifoldPlus (Huang et al., 2020).

**Quantitative Evaluation.** Considering the requirements of the text-guided texture synthesis task, we compare our method with previous methods in three dimensions: ❶ *Text matching*. We first evaluate how well the generated texture matches the text input, using the average CLIP score (Radford et al., 2021) across multiple views. ❷ *Quality*. Due to the lack of ground truth, we exploit an image-only perception metric CLIP-IQA (Wang et al., 2023) for assessing the quality, and calculate it from multiple views. ❸ *Consistency*. Since there is no previous work evaluating the multi-view consistency of 3D objects, we develop a metric called CLIP variance based on the idea that images of the same object from multiple views have the same semantics. Specifically, we use the CLIP visual encoder to extract features for multi-view rendering due to its ability to represent multiple semantics, and take the minimum value of cosine similarity between these features as the metric.

Tab. 1 shows the comparison with previiious state-of-the-art methods including Text2Mesh (Michel et al., 2022), Latent-Paint (Metzer et al., 2022), TEXTure (Richardson et al., 2023), and Text2Tex (Chen et al., 2023). For text matching, Text2Mesh achieves a significantly higher CLIP score than other methods since it directly using CLIP score as optimization objective. Our method get a comparable CLIP score to TEXTure, which demonstrates that the consistency guidance introduced in the multi-view image generation process does not hurt the text-to-image generation ability of 2D diffusion model. Moreover, our P3G significantly outperforms previous methods on CLIP-IQA. This is attributed to the consistency between different view that avoids obvious artifacts on the generated 3D texture, and high-quality images generated by 2D diffusion. As for the consistency, it is obvious that the optimization-based methods Text2Mesh and Latent-Paint work significantly well, at the cost of extremely low visual quality due to the generation capabilities of the model are not well released. This phenomenon also verifies the effectiveness of our idea that learning a pseudo 3D guidance by optimization where only semantics and layout are required. Overall, our P3G surpasses the counterpart TEXTure both in overall quality and consistency. It is also worth mentioning that Text2Tex achieves considerable consistency with P3G, because the textures it generates are ofen simple in color and detail, as shown by CLIP-IQA of Tab. 1 and Fig. 5.

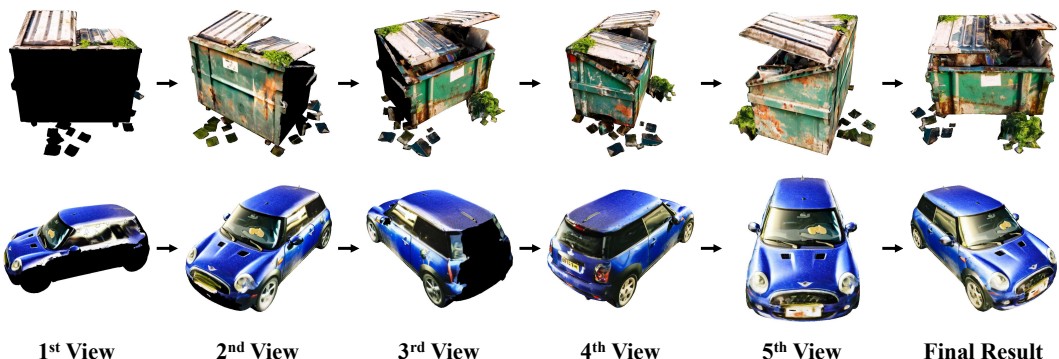

| | 1st View | 2nd View | 3rd View | 4th View | 5th View | Final Result |

Figure 7: Visualizations of the view selection strategy.

**User Study.** We compare our method with the inpainting-based approach TEXTure through user study. Other methods are omitted because the texture they generate are relatively simple . First, the participants are asked to answer whether there is obvious inconsistency in the generated texture. If the answer is yes, then the inconsistency index is increased by one. Then the user need to score the generated texture 0 or 1 based on their preference, and the final scores are added up as the overall quality indicator. Finally we calculate the average of these two indicators, and the results are shown in Tab. 2. Based on user feedback, our P3G is significantly better than TEXTure in both consistency and overall quality.

| | Inconsistency | Overall |
|---|---|---|
| TEXTure | 23.63 | 36.23 |
| P3G (*Ours*) | **15.55** | **42.45** |

Table 2: User study results.

**Qualitative Evaluation.** Fig. 5 shows some examples of the generated texture. Thanks to the generation ability of 2D diffusion model and the proposed pseudo 3D guidance for encouraging the consistency, our P3G can generate view-consistent and highly-detailed texture which also matches well with the input geometry and text.

## 4.3 ABLATION STUDY

**Effectiveness of Pseudo 3D Guidance.** For verifying the role of the pseudo 3D guidance on consistency by comparing with a well-designed inpainting-based method (*i.e.*, TEXTure) and a baseline which dose not incorporate the guidance when generating image. The results in Fig. 6 demonstrate that the pseudo 3D guidance successfully controls the randomness of generation thus producing consistent results in different part of the object, which is especially helpful for cylindrical objects that require extremely high consistency.

**View Selection Strategy.** We visualize the multi-view image generation process with the proposed view selection strategy in Fig. 7. With only negligible additional calculations, the view covering as wide an area as possible can be accurately estimated by our strategy, which finally speeds up the generation of the entire texture.

## 5 CONCLUSION

We present a novel method for high-quality text-driven 3D texturing. The automatic generation of 3D assets is an intriguing research topic and is of great value in many applications. While the pioneering works utilize the photorealistic 2D image generation ability of large-scale pre-trained generative models, the view inconsistency induced by the natural randomness of the generation process and the fact that 2D generative models are unaware of 3D consistency largely limits its performance. To move step further, we opt to learn a pseudo 3D guidance first and then use it to guide high-quality and view-consistent multi-view image generation. We propose a novel method for the learning of the pseudo 3D guidance based on the property that 2D rendering at any view of a 3D object obeys the real image distribution. Later, we design an efficient conditional generation pipeline that enables high-quality and view-consistent multi-view image generation according to the depth map, the learned pseudo 3D guidance, and the previously generated textures. We conduct both quantitative and qualitative evaluations on various 3D shapes and text descriptions. The experimental results demonstrate the superiority of the proposed method.

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

# A ADDITIONAL ABLATION STUDY

**Latent-to-RGB Optimization.** Fig. 8 shows the texture obtained by optimization in latent space and RGB space. Due to the small computational cost, the texture can quickly converge in the latent space, but the image resolution limits the sharpness of the texture. Taking it as initialization, the optimization in RGB space further improves the texture in detail by high-resolution images, finally producing precise guidance.

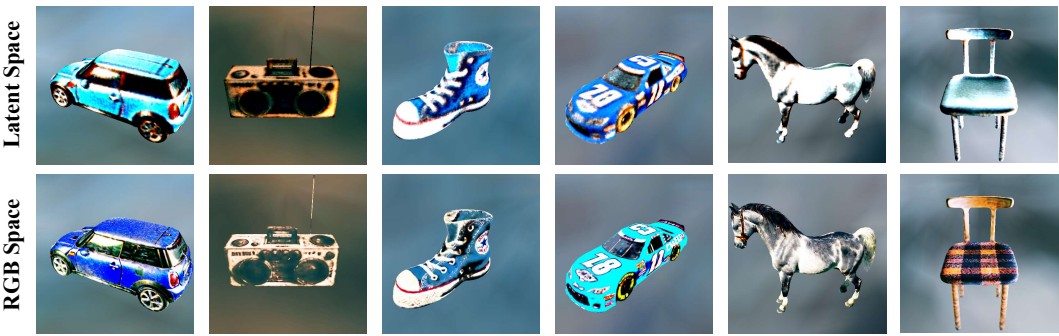

Figure 8: Visualizations of learned textures *w.r.t.* latent space and RGB space.

