# OpenReview forum: "Learning Pseudo 3D Guidance for View-consistent 3D Texturing with 2D Diffusion"
_ICLR.cc/2024/Conference — ICLR 2024 Conference Withdrawn Submission_

### Official Review · Reviewer_Nu8g · 2023-10-18

**Soundness:** 2 fair
**Presentation:** 3 good
**Contribution:** 2 fair
**Rating:** 5
**Confidence:** 4

**Summary:**

This article proposes a high-quality pseudo 3D guidance texture generation method, which is achieved through pseudo 3D guided learning and high-quality multi view image generation based on pseudo 3D guidance.

**Strengths:**

The most significant improvement of this article is combining SDS loss with depth-image diffusion for texture generation. And it maintains multi view consistency of textures through a series of technical methods. Compared to previous methods, it has achieved excellent results.

**Weaknesses:**

1. Perhaps it has improved the generation results compared to previous texture generation methods, but it is well known that the Dreamfusion-based works using SDS loss can generate remarkable geometry and textures simultaneously. What are the advantages of this method compared to these works?

2. The paper claims in the abstract that it is a text driven 3D texturing. So what is the effect of generating diverse style textures through text control? If possible, please provide a gallery.

3. Please explain the meaning of "pseudo".

**Questions:**

Please refer to weakness.

---

### Official Review · Reviewer_vaRt · 2023-10-28

**Soundness:** 2 fair
**Presentation:** 2 fair
**Contribution:** 2 fair
**Rating:** 5
**Confidence:** 3

**Summary:**

Given a mesh, this paper studies to texture it with text prompts.
It proposes a two-stage method: in the first stage, it generates coarse textures using a pseudo 3D guidance, i.e. depth-conditioned Stable diffusion.
In the second stage, it refines the existing textures to get high-resolution results with a view-adaptive textural completion, and view selection.
This method could give good results.

**Strengths:**

1.  In the related work section, this paper discusses some "old paper" ((De Bonet, 1997; Efros & Leung, 1999), which is good.
2. The results look good.

**Weaknesses:**

1. This paper doesn't provide useful text prompts for evaluating generated images.  In figure 6 and supp videos, there are only generated results without any text prompts. Consequently, it is very hard to evaluate whether the results respect the prompts or not since we don't know the text prompts.  In figure 5, the text prompts are not very helpful. For example, text prompts like "A radio” actually don't give many instructions for textures. Without text prompts or more detailed descriptions of styles or colors, these comparison becomes
 aesthetics comparison, and conclusions could be really subjective.  I understand some previous papers like Text2Tex use the same prompts and presentation style. But their baselines are quite blur or broken. And it is hard for me to see distinct qualitatively difference from the current presentation.
2. Some components are confusing. It is unclear how the proposed method is related to some previous methods. More specifically, could we regard "Pseudo 3D guidance " as "using depth conditioned stable diffusion in SDS"?  What's the difference between "View-adaptive Texture Completion, Efficient View Selection Strategy, View-Consistent Image Generation" compared to method of Text2Tex?
3. It would be very helpful if more methodology and results comparison to Text2Tex are provided, since I personally feel they are kind of similar.

**Questions:**

1. More comparisons to text2Tex, either in methodology or results perspective would be really helpful.
2. What's the benefits of optimising latent feature first and then switching to RGB features in the first stage, compared to directly optimising the RGB features?

---

### Official Review · Reviewer_ruEM · 2023-10-30

**Soundness:** 2 fair
**Presentation:** 2 fair
**Contribution:** 2 fair
**Rating:** 3
**Confidence:** 4

**Summary:**

This paper introduces a text-to-3D texturing method named Pseudo 3D Guidance (P3G). The authors propose to first learn a coarse but view-consistent texture by depth-guided score distillation sampling (SDS). They then use this coarse texture as semantics and layout guidance for generating high-quality view-consistent texture by the diffusion denoising process. The authors also introduces a view-adaptive texture completion to deal with overlapping part of the texture under different views and  an efficient view selection strategy to cover the texture with as few views as possible.

**Strengths:**

- The paper introduces P3G, which achieves 3D texturing via the learned Pseudo 3D Guidance at coarse stage. The proposed setup efficiently addresses the challenge of 3D texture consistency and improves the quality of the learned texture.
- The proposed view-adaptive texture completion and view selection strategy are feasible and simple.
- The proposed method superpasses exisiting methods in both overall quality and consistency of the generated texture.

**Weaknesses:**

- Limited novelty. Depth-conditioned diffusion model is well-used in previous methods, e.g., 3DFuse. The use of semantic and layout information is also not novel. For example, IDE-3D has applied them for 3D-aware face editing. This raises the question of whether the proposed network offers any distinct advantages or novel functionalities, such as editing capabilities, via the use of semantics and layout information. The so-called pseudo guidance appears to resemble a coarse-to-fine strategy, which is commonly employed in various applications. Additionally, it might be more reasonable to update the pseudo guidance in the second stage than making it locked. Conducting experiments to investigate this aspect would provide more valuable insights.
- Insufficient evaluations. The visualizations in Figure 5 do not clearly demonstrate the inconsistency problem present in previous methods, making it challenging to recognize the extent of the improvement achieved by the proposed approach. While the authors introduce a new view-selection strategy and a view-adaptive texture completion model (which are simple greedy algorithms), it is important to compare their performance against the strategy employed in TEXTure to understand their relative effectiveness. The quantitative numbers are not supportive.
- Insufficient ablation. No ablation has been considered for the design choices in (8) and (9).

**Questions:**

- In (7), should the smaller than operator be replaced by the greater than operator?
- In (9), it is not clear why setting the foreground always back to x^c for t < \tau would produce better results. Further explanation and analysis are needed to support this deisgn.

---

### Official Review · Reviewer_ij6J · 2023-11-01

**Soundness:** 2 fair
**Presentation:** 2 fair
**Contribution:** 2 fair
**Rating:** 3
**Confidence:** 4

**Summary:**

This paper proposes a 3D guidance on high-level semantics and layout of the texture. Through this guidance, most of the randomness in high-level semantics and layout can be avoided and the remaining randomness in local details can be largely controlled by inpainting, therefore ensuring a global consistency of the generated 3D texture when observed from different views. The presentation and experiments of the paper need to be improved.

**Strengths:**

- This paper proposes a pseudo 3D guidance learning module, which can serve as layout and semantics guidance of the final image.
- This paper introduces the inpainting operation to avoid the inconsistency in local details induced by the diffusion denosing process.
- The qualitative results on some 3D shapes outperform other compared methods.

**Weaknesses:**

- The contributions of this paper are unclear.
- The text description of 3D mesh is too simple, can you show some results on the more complex texts?
- This paper only conducts experiments on 50 multi-category meshes collected from ModelNet and ShapeNet. I want to see more experimental results on the Objaverse datasets.
- In Table 1, the CLIP Score and CLIP Variance are worse than other compared methods. Although the consistency is improved, I am concerned about the quality of the 3d textures
- In Figure 6, the corresponding text is not provided.

**Questions:**

- In Eq. 2, what does $c$ mean?
- Given a text and 3D mash, how long does it take to generate high-quality and view-consistent 3d textures?